

# Genomics analysis of genes encoding respiratory burst oxidase homologs (RBOHs) in jatropha and the comparison with castor bean

Yongguo Zhao[1,2] and Zhi Zou[2]

[1] Guangdong University of Petrochemical Technology, Maoming, Guangdong, China
[2] Hainan Key Laboratory for Biosafety Monitoring and Molecular Breeding in Off-Season Reproduction Regions, Key Laboratory of Biology and Genetic Resources of Tropical Crops, Ministry of Agriculture and Rural Affairs, Institute of Tropical Biosciences and Biotechnology, Chinese Academy of Tropical Agricultural Sciences, Haikou, Hainan, China

Corresponding author
Zhi Zou, zouzhi@itbb.org.cn, zouzhi2008@126.com

## ABSTRACT

Respiratory burst oxidase homologs (RBOHs), which catalyze the production of superoxide from oxygen and NADPH, play key roles in plant growth and development, hormone signaling, and stress responses. Compared with extensive studies in model plants arabidopsis and rice, little is known about RBOHs in other species. This study presents a genome-wide analysis of *Rboh* family genes in jatropha (*Jatropha curcas*) as well as the comparison with castor bean (*Ricinus communis*), another economically important non-food oilseed crop of the Euphorbiaceae family. The family number of seven members identified from the jatropha genome is equal to that present in castor bean, and further phylogenetic analysis assigned these genes into seven groups named RBOHD, -C, -B, -E, -F, -N, and -H. In contrast to a high number of paralogs present in arabidopsis and rice that experienced several rounds of recent whole-genome duplications, no duplicate was identified in both jatropha and castor bean. Conserved synteny and one-to-one orthologous relationship were observed between jatropha and castor bean *Rboh* genes. Although exon-intron structures are usually highly conserved between orthologs, loss of certain introns was observed for *JcRbohB*, *JcRbohD*, and *RcRbohN*, supporting their divergence. Global gene expression profiling revealed diverse patterns of *JcRbohs* over various tissues. Moreover, expression patterns of *JcRbohs* during flower development as well as various stresses were also investigated. These findings will not only improve our knowledge on species-specific evolution of the *Rboh* gene family, but also provide valuable information for further functional analysis of *Rboh* genes in jatropha.

## INTRODUCTION

Respiratory burst oxidase, first identified in human phagocytic cells, is a key enzyme that catalyzes the production of superoxide from oxygen and NADPH (*Sumimoto, 2008*). In higher plants, this enzyme is known as respiratory burst oxidase homolog

(RBOH), which was characterized with the presence of one N-terminal NADPH_Ox domain, one Ferric reductase like transmembrane component, one FAD-binding domain, one NAD-binding domain, and several calcium-binding EF-hand motifs (*Sumimoto, 2008*; *Kaur et al., 2018*). Since the first RBOH was characterized in rice (*Oryza sativa*), a growing number of homologs have been identified in plant lineages (*Groom et al., 1996*; *Kaur et al., 2014*). Genome-wide surveys have also been performed in several species, e.g., seven members from grape (*Vitis vinifera*), seven members from strawberry (*Fragaria ×ananassa*), nine members from rice (*Oryza sativa*), nine members from cassava (*Manihot esculenta*), ten members from arabidopsis (*Arabidopsis thaliana*), ten members from rubber (*Hevea brasiliensis*), and 15 members from wheat (*Triticum aestivum*) (*Sagi & Fluhr, 2006*; *Cheng et al., 2013*; *Wang et al., 2013*; *Hu et al., 2018*; *Zhang et al., 2018*; *Zou & Yang, 2019a*). Evidence showed that RBOHs play key roles in growth, development, hormone signaling, and stress responses of plants (*Marino et al., 2012*; *Kaur et al., 2014*; *Angelos & Brandizzi, 2018*; *Wang et al., 2018*; *Zou & Yang, 2019a*). Taking the well-studied model plant arabidopsis for example, *AtRbohD* and *AtRbohF* function not only in disease resistance (*Chaouch, Queval & Noctor, 2012*) and salt stress tolerance (*Xie et al., 2011*), but also are essential for jasmonic acid-induced expression of genes regulated by the MYC2 transcription factor (*Maruta et al., 2011*). *AtRbohD* also participates in endosperm development (*Penfield et al., 2006*) and ABA-mediated ROS production and stomatal closure (*Zhang et al., 2009*). Although the activation mechanism of AtRBOHD and AtRBOHF is similar in stress responses, AtRBOHD was shown to have a significantly greater ROS-producing activity than AtRBOHF (*Kimura et al., 2013*). Analysis of mutants also revealed that *AtRbohD* is the major constitutively active member, and *AtRbohF* is a biotic stress-inducible member (*Torres, Dangl & Jones, 2002*). *AtRbohB* participates in seed after-ripening (*Müller et al., 2009*) and *AtRbohC* functions in root-hair-tip growth (*Takeda et al., 2008*). *AtRbohJ* and *AtRbohH* play essential roles in pollen tube tip growth *via* $Ca^{2+}$-activated ROS production (*Kaya et al., 2014*), whereas *AtRbohJ* also functions in salt stress tolerance (*Evans et al., 2005*). Compared with extensive studies in arabidopsis, research in other plants is still in its infancy.

Jatropha (also known as physic nut, *Jatropha curcas* L., $2n = 22$) and castor bean (*Ricinus communis* L., $2n = 20$) are two economically important non-food oilseed crops of the Euphorbiaceae family (*Zou et al., 2015b*; *Zou et al., 2018*; *Zou, 2018*; *Zou & Zhang, 2019*). Jatropha is characterized as a small tree native to central America, whereas castor bean is a perennial shrub originated from Africa (*Zou et al., 2016a*; *Zou et al., 2016b*; *Zou et al., 2018*). Based on available genome sequences, the number of 27,172 putative protein-coding genes in jatropha is relatively less than 31,221 in castor bean, which is consistent with a smaller genome size of 320 Mb in jatropha than 400 Mb in castor bean (*Chan et al., 2010*; *Wu et al., 2015*). According to comparative genomics analysis, these two species didn't experience additional whole-genome duplication (WGD) after the so-called whole-genome triplication gamma (γ) event occurred at approximately 117 Mya (million years ago) (*Chan et al., 2010*; *Jiao et al., 2012*; *Wu et al., 2015*). Further analysis revealed that these two species may diverge from a common ancestor at about 49.4 Mya (*Wu et al., 2015*). Despite the importance of RBOHs, little information is available in these two

particular species. Herein, we present a genome-wide comparative analysis of the *Rboh* gene family in jatropha and castor bean, including gene structures, evolutionary relationships, motif distribution, as well as expression profiles with a focus on jatropha. Our findings provide valuable information for further functional analysis of *Rboh* genes in these two species.

## MATERIALS & METHODS

### Identification and manual curation of *Rboh* family genes

Arabidopsis and rice *Rboh* genes described before were retrieved from TAIR10 (*Lamesch et al., 2012*) or RGAP7 (*Sakai et al., 2013*), respectively, and their accession numbers are shown Table S1. At/OsRBOH proteins were used as queries to search for homologs from jatropha and castor bean genomes that were downloaded from NCBI (*NCBI Resource Coordinators, 2018*) or Phytozome v12 (*Goodstein et al., 2012*), respectively. Sequences with an *E*-value of less than 1E-5 in the tBLASTn search (*Altschul et al., 1997*) were collected and computationally predicted gene models were manually revised with cDNAs, ESTs (expressed sequence tags) and RNA-seq (RNA sequencing) reads that are available in NCBI (last accessed Jan 2018). Presence of the conserved NADPH_Ox domain in candidate RBOHs was confirmed using MOTIF Search (http://www.genome.jp/tools/motif/) and gene structures were displayed using GSDS (*Hu et al., 2015*). Homology search for nucleotides or ESTs and expression annotation using RNA-seq data were performed as previously described (*Zou et al., 2015a*; *Zou et al., 2015b*).

### Synteny analysis, sequence alignment and phylogenetic analysis

Synteny analysis and multiple sequence alignment of deduced RBOH proteins were carried out as previously described (*Zou, Xie & Yang, 2017*; *Zou, Zhu & Zhang, 2019b*). Phylogenetic trees were constructed using MEGA (version 6.0) (*Tamura et al., 2013*), implementing the maximum likelihood method with a bootstrap of 1,000 replicates. Orthologs were determined using the BRH (Best Reciprocal Hit) method as described before (*Zou et al., 2018*) and further confirmed using the result from synteny analysis of jatropha and castor bean. Sequence alignment of Jc/RcRBOHs was displayed using Boxshade (http://sourceforge.net/projects/boxshade/).

### Protein properties and conserved motif analysis

Various physical and chemical parameters of RBOH proteins were calculated using ProtParam (*Gasteiger et al., 2005*) and subcellular localization was predicted using Plant-mPLoc (http://www.csbio.sjtu.edu.cn/bioinf/plant/). Analysis of conserved motifs was performed using MEME (http://meme-suite.org/tools/meme): any number of repetitions distributed in sequences, the maximum number of 15 motifs, and the width of each motif ranging from six to 180 residues.

### Gene expression analysis

Various transcriptome data were retrieved from NCBI SRA and detailed information is shown in Table S2. Raw reads were first filtered using fastQC (http://www.bioinformatics.

babraham.ac.uk/projects/fastqc/), and resulted clean reads were mapped to the coding sequences (CDS) of revised *JcRbohs* or *RcRbohs* as well as other protein-coding genes using Bowtie 2 (*Langmead & Salzberg, 2012*), and the relative transcript level was represented by FPKM (fragments per kilobase of exon per million fragments mapped, for pair-ended samples) or RPKM (Reads per kilobase per million mapped reads, for single-ended samples) (*Mortazavi et al., 2008*; *Trapnell et al., 2010*). The significance of gene expression difference was determined with parameters "FDR < 0.001" and "log2Ratio ≥ 1" with our in-house scripts. Unless specific statements, the tools used in this study were performed with default parameters.

## RESULTS

### Characterization of seven *Rboh* family genes in jatropha

A survey of the jatropha genome resulted in seven loci that were proven to encode *Rboh* genes. Considering the extensively functional analysis performed in arabidopsis, *JcRbohB*, *JcRbohC*, *JcRbohD*, *JcRbohE*, *JcRbohF*, and *JcRbohH* were named after their best orthologs in arabidopsis, whereas *JcRbohN* represents a novel group member with no ortholog in arabidopsis as well as rice (see below). Based on the result of mining RNA-seq data, all these genes were shown to be expressed. Moreover, *JcRbohC*, *JcRbohD*, and *JcRbohF* have corresponding ESTs in NCBI GenBank (Table 1). Compared with the original genome annotation, one gene model was manually optimized on the basis of read alignment: the locus JCGZ_22480 (*JcRbohN*) was predicted to contain 14 introns putatively encoding 755 residues (KDP26234) and actually it only contains 13 introns encoding 736 residues (see File S1). Based on available genetic markers (*Wu et al., 2015*), six *Rboh*-encoding scaffolds were further anchored to five out of the 11 chromosomes (Chrs) (Fig. 1).

### Characterization of seven *Rboh* family genes in castor bean

To facilitate evolutionary analysis, *Rboh* family genes present in castor bean were also identified, resulting in seven genes that are also distributed across six scaffolds. *RcRbohB* and *RcRbohD* are located on the same scaffold, which is similar to *JcRbohB* and *JcRbohD* as observed in jatropha (Table 1), corresponding to the close phylogenetic relationship between these two Euphorbiaceous plants. A significant level of syntenic relation and one-to-one orthologous relationship were observed between jatropha and castor bean *Rboh* genes, supporting their divergence before speciation of these two species. As shown in Fig. 1, synteny analysis also allows anchoring all *RcRbohs* to five jatropha chromosomes. Based on manual curation, three of the computationally predicted gene models were optimized. The locus 30147.t000648 (*RcRbohC*) was predicted to encode 710 residues (30147.m014377), and it actually represents only the 3′ sequence of the gene which encodes 913 residues (see File S2). The locus 30128.t000051 (*RcRbohD*) was predicted to harbor 10 introns putatively encoding 709 residues (30128.m008590), which is relatively shorter than its ortholog in jatropha. In fact, *RcRbohD* was also proven to harbor 11 introns, though it contains at least two alternative splicing isoforms. The most common isoform encodes 916 residues (see File S3), whereas the short isoform (i.e., 30128.m008590) is derived from the skipping of the tenth intron. The locus 29941.t000003 (*RcRbohN*) was predicted to harbor 12 introns

**Table 1** *Rboh* family genes identified in this study.

| Gene name | Locus name | Position | Nucleotide length (bp, from start to stop codons) | | Intron no. | EST no. | AA | MW (kDa) | *p* I | GRAVY | TMH | Ortholog | |
|---|---|---|---|---|---|---|---|---|---|---|---|---|---|
| | | | CDS | Gene | | | | | | | | At | Os |
| *JcRbohC* | JCGZ_26338 | scaffold906:2548368-2554294 | 2715 | 5349 | 11 | 5 | 904 | 102.57 | 9.13 | −0.305 | 4 | *AtRbohC* *AtRbohA* *AtRbohG* | – |
| *JcRbohD* | JCGZ_23064 | scaffold779:96300-100604 | 2772 | 3819 | 10 | 1 | 923 | 103.85 | 9.20 | −0.286 | 4 | *AtRbohD* | *OsRbohI* |
| *JcRbohB* | JCGZ_23317 | scaffold779:1441569-1437068 | 2676 | 3882 | 10 | 0 | 891 | 101.36 | 9.07 | −0.285 | 4 | *AtRbohB* | *OsRbohB* *OsRbohH* |
| *JcRbohE* | JCGZ_07528 | scaffold211:3210360-3215422 | 2772 | 4858 | 13 | 0 | 923 | 104.39 | 8.91 | −0.187 | 4 | *AtRbohE* | *OsRbohF* *OsRbohG* |
| *JcRbohF* | JCGZ_20527 | scaffold660:1101645-8731 | 2862 | 7448 | 13 | 1 | 953 | 108.19 | 9.38 | −0.264 | 4 | *AtRbohF* *AtRbohI* | *OsRbohA* *OsRbohC* |
| *JcRbohN* | JCGZ_22480 | scaffold729:242504-247775 | 2211 | 5272 | 13 | 0 | 736 | 84.31 | 9.35 | −0.089 | 4 | – | – |
| *JcRbohH* | JCGZ_05621 | scaffold18:2574685-2569687 | 2649 | 4999 | 13 | 0 | 882 | 100.37 | 9.44 | −0.227 | 6 | *AtRbohH* *AtRbohJ* | *OsRbohE* *OsRbohD* |
| *RcRbohC* | 30147.t000648 | scaffold30147:1591736-1597026 | 2742 | 4658 | 11 | 0 | 913 | 103.59 | 9.04 | −0.362 | 4 | *AtRbohC* *AtRbohA* *AtRbohG* | – |
| *RcRbohD* | 30128.t000051 | scaffold30128:2636281-2631846 | 2751 | 3885 | 11 | 3 | 916 | 103.17 | 9.04 | −0.257 | 4 | *AtRbohD* | *OsRbohI* |
| *RcRbohB* | 30128.t000279 | scaffold30128:1185430-1190626 | 2667 | 4573 | 11 | 0 | 888 | 101.47 | 9.21 | −0.334 | 4 | *AtRbohB* | *OsRbohB* *OsRbohH* |
| *RcRbohE* | 29739.t000141 | scaffold29739:892440-898120 | 2805 | 5681 | 13 | 0 | 934 | 105.82 | 8.78 | −0.239 | 4 | *AtRbohE* | *OsRbohF* *OsRbohG* |
| *RcRbohF* | 30190.t000521 | scaffold30190:3095171-3103214 | 2823 | 6880 | 13 | 0 | 940 | 106.96 | 9.32 | −0.259 | 4 | *AtRbohF* *AtRbohI* | *OsRbohA* *OsRbohC* |
| *RcRbohN* | 29941.t000003 | scaffold29941:28891-17709 | 2196 | 6347 | 12 | 0 | 731 | 83.76 | 9.26 | −0.122 | 5 | – | – |
| *RcRbohH* | 30039.t000013 | scaffold30039:82700-87109 | 2664 | 4410 | 13 | 0 | 887 | 101.22 | 9.13 | −0.209 | 6 | *AtRbohH* *AtRbohJ* | *OsRbohE* *OsRbohD* |

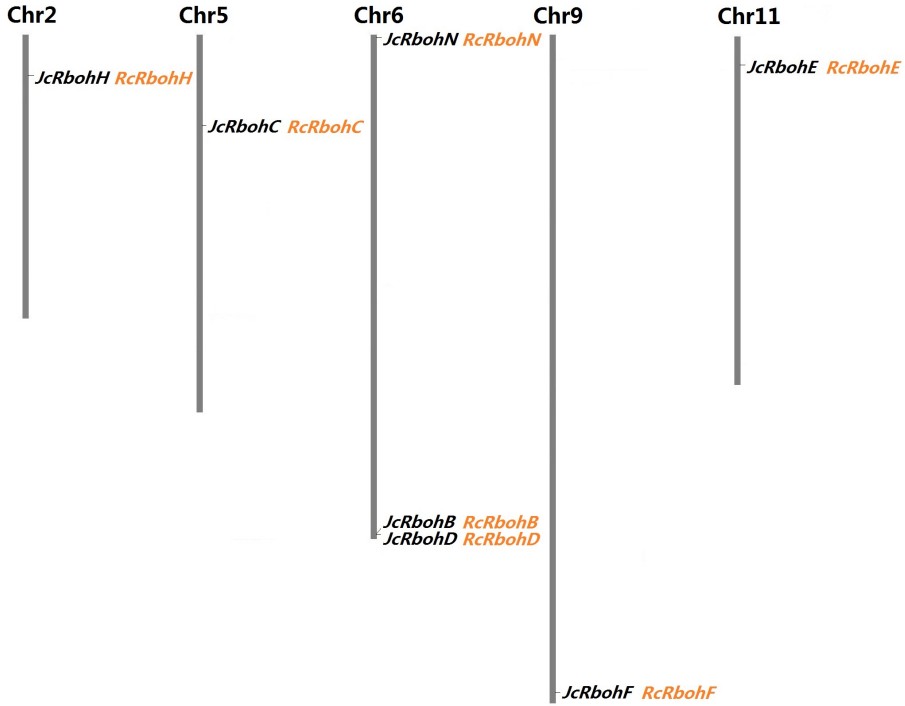

**Figure 1   Chromosomal locations of seven *JcRboh* genes and their collinear genes in castor bean.** Five *Rboh*-encoding chromosomes and the chromosome serial number are indicated at the top. *RcRboh* genes shown just behind their collinear genes in jatropha are marked in orange.

putatively encoding 666 residues (29941.m000220), which is relatively shorter than its ortholog in jatropha. Actually, read alignment indicated that this gene contains 13 introns encoding 731 residues (see File S4).

## Phylogenetic analysis, exon-intron structures, and conserved motifs

To reveal the evolutionary relationship of *Rboh* family genes and facilitate the transfer of functional information obtained in model plants, an unrooted phylogenetic tree was constructed from full-length Jc/Rc/At/OsRBOHs, where arabidopsis and rice represent the model eudicot or monocot, respectively. As shown in Fig. 2A, these RBOHs were clustered into seven groups named RBOHC, -D, -B, -E, -F, -N, and -H, which is highly consistent with the BRH-based homologous analysis (Table 1). As for both jatropha and castor bean, a single member was found in each group. By contrast, recent duplicates were found in most groups of arabidopsis and rice *Rboh* gene families, which were shown to result from WGD as well as local duplication (see Table S1). However, RBOHC is absent from rice, whereas RBOHN is absent from both arabidopsis and rice, suggesting lineage-specific gene loss. The classification is further supported by exon-intron structure patterns as well as conserved motifs. As shown in Fig. 2B, RBOHB, RBOHC, and RBOHD feature 11 introns, whereas RBOHF, RBOHE, RBOHH, and RBOHN usually contain 13 introns. However, gene-specific gain or loss of certain introns was also observed, i.e., *JcRbohB*, *JcRbohD*, *RcRbohN* as well as *AtRbohC*, *AtRbohG*, *AtRbohD*, *AtRbohH*, *AtRbohJ*, *OsRbohH*,

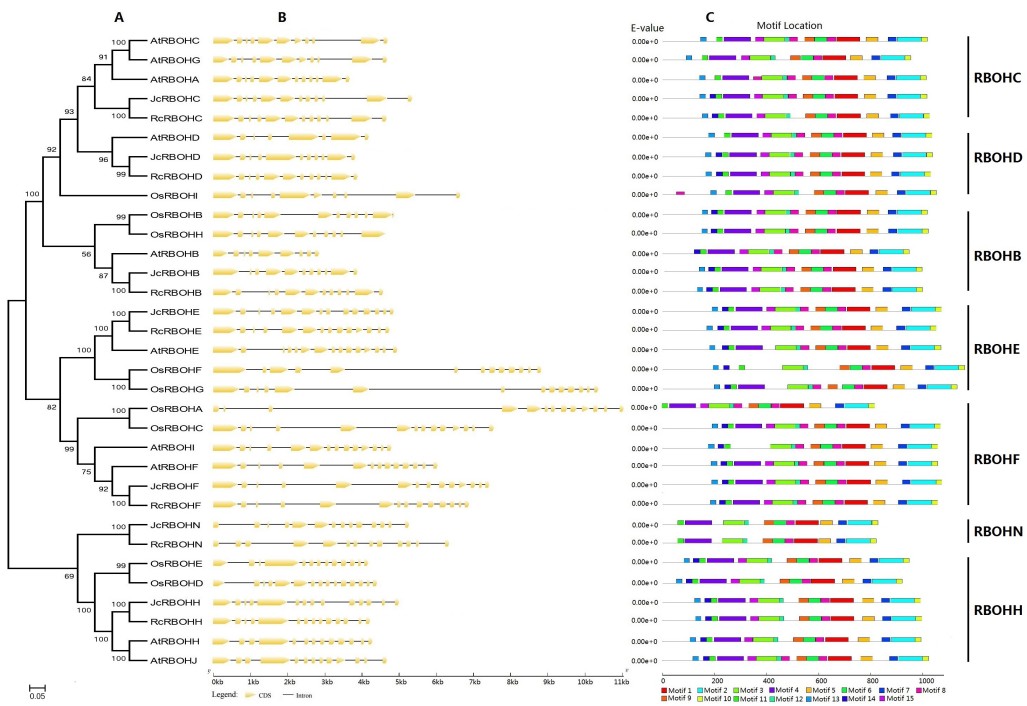

**Figure 2 Structural and phylogenetic analysis of *Rboh* family genes in jatropha, castor bean, arabidopsis, and rice.** (A) The unrooted phylogenetic tree resulting from full-length RBOH proteins with MEGA6, where the distance scale denotes the number of amino acid substitutions per site and bootstrap values are indicated on the top of each clade. (B) The graphic representation of exon-intron structures displayed using GSDS. (C) The distribution of conserved motifs among RBOH proteins, where different motifs are represented by different color blocks as indicated at the bottom of the figure and the same color block in different proteins indicates a certain motif.

*OsRbohA*, and *OsRbohD*. The variety of exon-intron structure in both jatropha and castor bean seems to be considerably less frequent than that in arabidopsis and rice. Compared with exon, the intron length of *Rboh* genes examined in this study is relatively more variable, where the third intron of *OsRbohA* harbors the maximum of 6,173 bp (Fig. 2B).

Several parameters of deduced RBOH proteins were also computed as shown in Table 1. JcRBOHs and RcRBOHs consist of 736–953 or 731–940 amino acids (AA), respectively, and the average sequence length of 887 AA in these two species is relatively smaller than 906 AA in arabidopsis or 903 AA in rice. The average theoretical MW (molecular weight) of 100.72 kDa in jatropha and 100.86 kDa in castor bean is similar to 101.55 kDa in rice or 102.93 kDa in arabidopsis. The average p*I* (isoelectric point) value of 9.21 in jatropha and 9.11 in castor bean is also similar to 9.20 in arabidopsis or 9.34 in rice. The GRAVY (grand average of hydropathicity) value varies from −0.081 to −0.362, supporting their hydrophilic feature (Table 1 and Table S1 ). Subcellular localization analysis showed that all examined RBOHs are located to the cell membrane. In addition to six transmembrane α-helices containing two hemes, several typical domains such as Ferric_reduct, FAD_binding_8, NAD_binding_6, NADPH_Ox, and EF-hand motif were found in all Jc/RcRBOHs, though

Jc/RcRBOHN lack a large part of the N-terminal as observed in other proteins (Fig. 3). In fact, the N-terminal of RBOHs was shown to be relatively variable related to the C-terminal.

Conserved motifs were further analyzed by using MEME as shown in Fig. 2C and Fig. S1. Among 15 motifs identified, Motifs 1–3, 5–12 are broadly distributed; Motif 4 is only absent from AtRBOHI and OsRBOHF, and Motifs 13–15 are absent from JcRBOHN and RcRBOHN. Moreover, Motif 13 is also absent from AtRBOHB and OsRBOHA; Motif 14 is also absent from AtRBOHC, AtRBOHA, AtRBOHG, AtRBOHD, OsRBOHD, and OsRBOHA; and, Motif 15 is also absent from AtRBOHA, AtRBOHI, AtRBOHE, OsRBOHF, and OsRBOHG. Motifs 13, 14, and 11 are part of the NADPH_Ox domain, which was proven to catalyze superoxide production; Motif 4 includes two EF-hand motifs, which are related to $Ca^{2+}$-binding and $Ca^{2+}$-dependent phosphorylation; Motifs 3, 12, 8, 9, and 6 are part of the Ferric_reduct domain that is similar to ferric reductase; Motif 1 is characterized as the FAD_binding_8 domain that bears the FAD-binding site; and, Motifs 5, 7, 2, and 10 are part of the NAD_binding_6 domain that bears the NAD-binding site (Fig. 2C).

## Expression profiles of *JcRboh* genes

Despite the expression of all *JcRbohs*, the transcript of *JcRbohH* was only detected in late stages of ovule and stamen development as well as a mixed sample of root, mature leaf, flower, developing seed, and embryo (*Natarajan & Parani, 2011*; *Hui et al., 2017*), and the transcript level was shown to be extremely low. A global view of *JcRboh* expression profiles was further investigated based on transcriptomes of several typical tissues, i.e., roots from 15-day-old seedlings, half expanded and mature leaves from 4-year-old plants, developing seeds from fruits harvested 19–28 days after pollination (DAP), undifferentiated inflorescence of 0.5 cm diameter (IND), female flowers with carpel primordia beginning to differentiate (PID1), female flowers with three distinct carpels formed (PID2), male flowers with stamen primordia beginning to differentiate (STD1), and male flowers with ten complete stamens formed (STD2). Results showed that the total family transcripts are most abundant in PID1 (defined as Class I); moderate in IND, seed, STD2, PID2, root, and STD1 (defined as Class II, accounting for 36–59% of Class I); and, relatively low in mature leaf and leafage (defined as Class III, accounting for 21–27% of Class I). *JcRbohC* and *JcRbohD* are constitutively expressed in most tissues examined: *JcRbohC* occupies 72%, 63%, 46%, 25%, 23% or 23% of the total transcripts in seed, mature leaf, root, leafage, and IND, respectively; and, *JcRbohD* occupies 62%, 50%, 49%, 42%, 38% or 24% in STD2, PID1, STD1, PID2, INDs, and root, respectively. Additionally, *JcRbohF* occupies 68%, 27% or 21% of the total transcripts in leafage, mature leaf, and root, respectively, though its transcript level is extremely low in IND, PID1, PID2, STD1, and STD2. Compared with leafage, *JcRbohC* is upregulated by about 3.0 folds in mature leaf, whereas *JcRbohF* is downregulated by about 2.3 folds. Compared with IND, *JcRbohD* is upregulated by about 2.2 folds in PID1; *JcRbohC* is downregulated by about 2.3 folds in PID2; *JcRbohB* and *JcRbohC* are downregulated by about 2.9 or 2.7 folds in STD1, respectively; and, *JcRbohC* is downregulated by about 2.2 folds in STD2. Compared with PID1, there are 0.4, 0.5 or 0.4 fold-changes in PID2 for *JcRbohB*, *JcRbohC*, and *JcRbohD*, respectively. According to their expression patterns, seven *JcRbohs* could be classed into four main clusters: Cluster

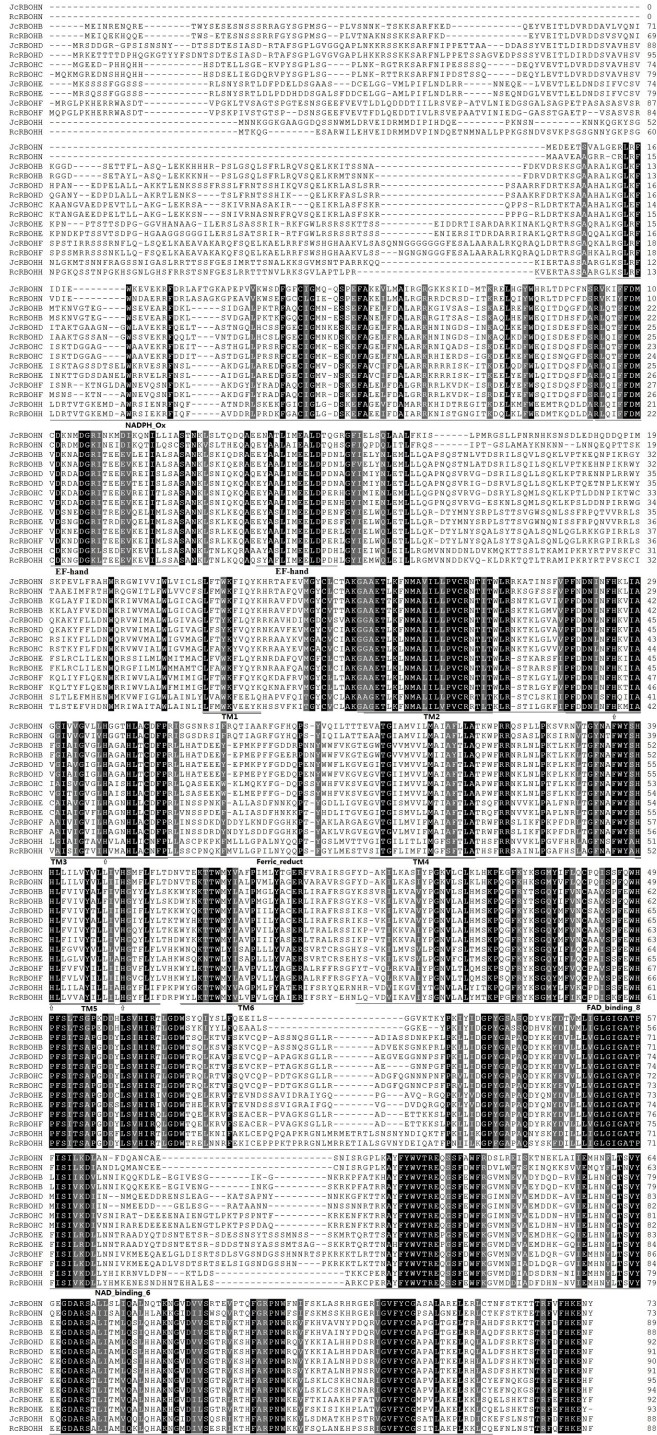

**Figure 3** **Sequence alignment of jatropha and castor bean RBOH proteins.** Identical and similar amino acids are highlighted in black or dark grey, respectively. Histidine residues involved in heme binding are indicated by arrows. Conserved domains such as NADPH_Ox, EF-hand motif, Ferric_reduct, FAD_binding_8, and NAD_binding_6 are indicated by single underlines, whereas putative transmembrane α-helices (TMs) are indicated by double underlines.

I is predominantly expressed in PID1, including *JcRbohB*, *JcRbohD*, and *JcRbohN*; Cluster II which includes *JcRbohC* is preferentially expressed in seed; Cluster III which includes *JcRbohF* is typically expressed in leafage; and Cluster IV which includes *JcRbohE* is mostly expressed in root (Fig. 4).

The response to drought and salt stresses was investigated in leaves and roots of 8-week old seedlings as described before (*Zhang et al., 2014*; *Zhang et al., 2015*). After withholding irrigation for 1, 4 or 7 d, all *JcRbohs* were shown to be significantly regulated at least one time point in at least one of the examined tissues. For 1 d, about 3.8-folds downregulation was observed for *JcRbohD* in leaf, whereas in root, decrease of about 2.6, 3.9, and 2.2 folds was observed for *JcRbohC*, *JcRbohN* or *JcRbohE*, respectively. For 4 d, about 4.8-folds decrease was observed for *JcRbohN* in root, while in leaf, increase of about 2.8 and 23.4 folds was observed for *JcRbohB* or *JcRbohN*, respectively. For 7 d, decrease of 2.0 and 2.2 folds was observed for *JcRbohD* or *JcRbohF* in root, respectively; by contrast, in leaf, increase of about 3.4, 10.7, 399.1 and 5.9 folds was observed for *JcRbohB*, *JcRbohD*, *JcRbohN* or *JcRbohE*, respectively. After the treatment with 100 mM NaCl for 2 h, 2 d or 7 d, two genes were significantly regulated (i.e., *JcRbohD* and *JcRbohN*): for 2 h, downregulation of *JcRbohD* (5.1 folds) and *JcRbohN* (7.8 folds) was observed in leaf or root, respectively; and upregulation of *JcRbohN* in both leaf and root was observed at later two time points, i.e., 5.3 and 12.3 folds for 2 d in leaf or root, respectively; and, 8.6 and 8.1 folds for 7 d in leaf or root, respectively (Fig. 5A).

The response to *Colletotrichum gloeosporioides* in leaf was analyzed on the basis of transcriptomes of two different genotypes, i.e., the susceptible RJ127 and the resistant 9-1. The result showed that *JcRbohN* was significantly upregulated in leaves of both the susceptible RJ127 and the resistant 9-1, though the fold-change is highly distinct (i.e., 10.4 *vs* 2.9 folds) (Fig. 5B). Additionally, in the case of RJ127, *JcRbohD* was significantly induced, whereas *JcRbohF* was inhibited (Fig. 5B), implying different regulation mechanism of these two cultivars.

Regulation of *JcRbohs* by hormones such as gibberellin acid (GA) and 6-benzylaminopurine (BA) (*Ni et al., 2017*) was also investigated. After application of 10 μM BA to young axillary buds for 12 h, the expression of *JcRbohN* increased by about 9.5 folds, by contrast, no such effect was observed for the same concentration of GA (Fig. 5B).

## DISCUSSION

Despite the crucial role of RBOHs in various plant processes (*Kaur et al., 2014*), little information is available in jatropha and castor bean, two Euphorbiaceous plants of economic importance. The availability of their genome sequences and various transcriptome datasets allows us to analyze this special gene family from a global view. In this study, a genome-wide identification and manual curation were performed, which results in seven *Rboh* family genes from both species. The family number is same as that present in grape and strawberry, but relatively smaller than other angiosperm plants reported thus far (*Sagi & Fluhr, 2006*; *Cheng et al., 2013*; *Wang et al., 2013*; *Chang et al., 2016*; *Hu et al., 2018*; *Zhang et al., 2018*; *Zou & Yang, 2019a*). The result is consistent with

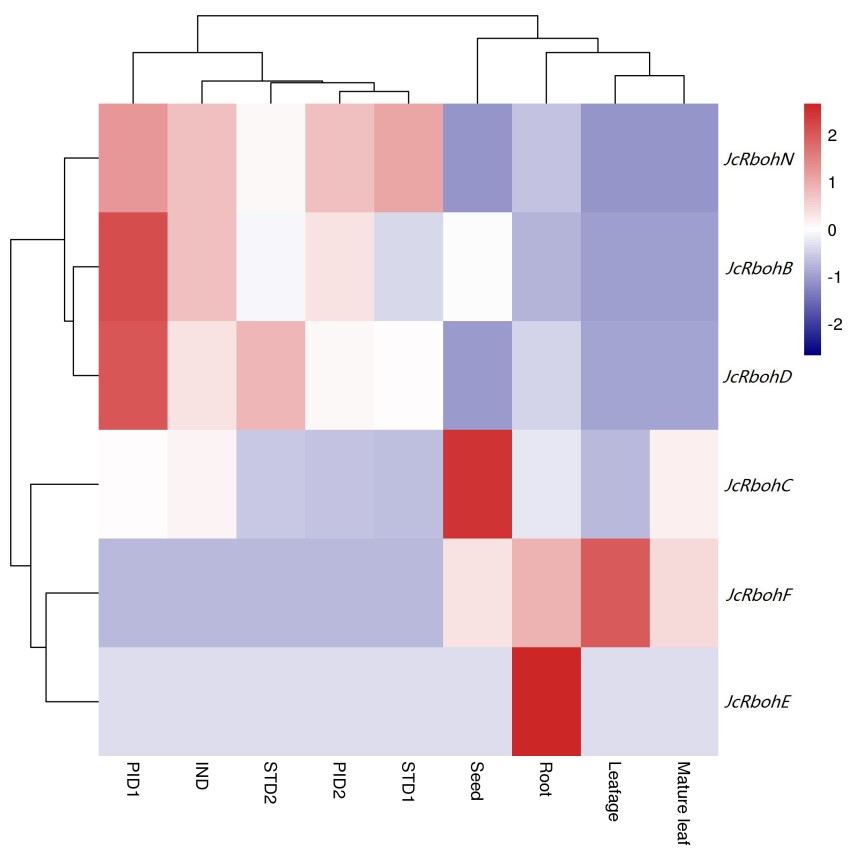

**Figure 4** **Expression profiles of *JcRboh* genes in various tissues or developmental stages.** Color scale represents FPKM normalized $\log_{10}$ transformed counts, where navy indicates low expression and fire-brick3 indicates high expression. (IND: Undifferentiated inflorescence of 0.5 cm diameter; PID1: female flower with carpel primordia beginning to differentiate; PID2: female flower with three distinct carpels formed; STD1: male flower with stamen primordia beginning to differentiate; STD2: male flower with ten complete stamens formed; Leafage: half expanded leaves; Leaf: mature leaves that have fully expanded).

the fact that both jatropha and castor bean didn't experience any recent WGD, which act as a main force for expansion of genes or gene families (*Qiao et al., 2019*; *Zou et al., 2016a*; *Zou, Xie & Yang, 2017*; *Zou, Yang & Zhang, 2019a*; *Zou, Zhu & Zhang, 2019b*; *Zou & Yang, 2019a*; *Zou & Yang, 2019b*; *Zou & Yang, 2019c*).

Phylogenetic and BRH-based analyses were adopted to divide these genes into seven groups, i.e., RBOHD, -C, -B, -E, -F, -H, and -N. The former six groups are also present in arabidopsis and/or rice, whereas RBOHN represents a recently reported group (*Zou & Yang, 2019a*). When taking advantage of *Os/At/Jc/RcRbohs* to analyze their orthologs in genome-sequenced plants available in Phytozome and other public databases, we found that RBOHE is relatively primitive, which is also present in *Physcomitrella patens*, a basal lineage of land plants without well-developed vasculature (*Rensing et al., 2008*). RBOHD is more likely to first appear in ancient seed plants, since it is present in gymnosperms but not in *Selaginella moellendorffii*, a member of an ancient vascular lineage (*Banks et al., 2011*). Both RBOHH and RBOHN can be traced back to *Amborella trichopoda*, a basal angiosperm plant
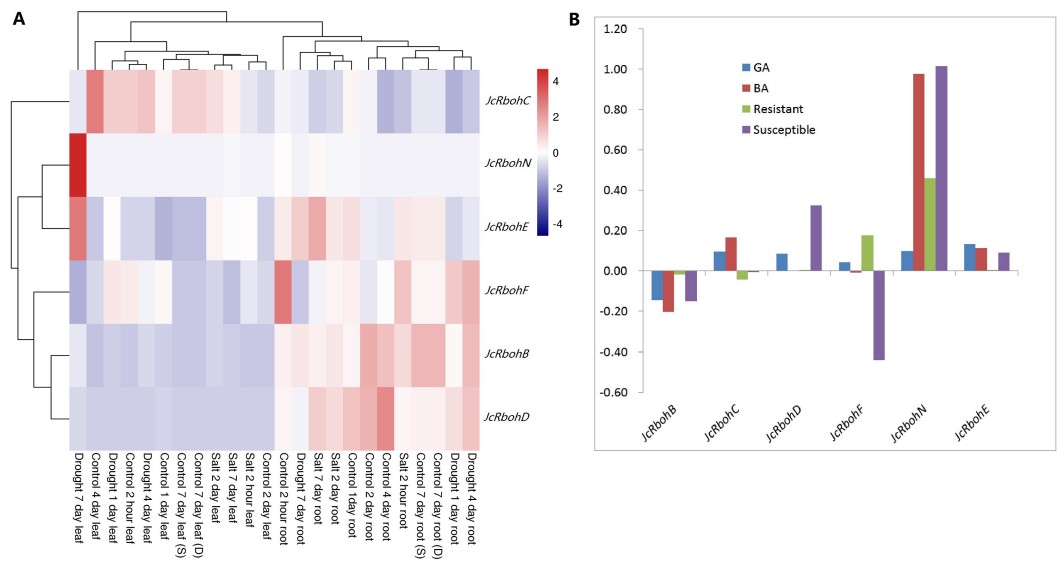

**Figure 5** **Expression profiles of *JcRboh* genes upon various stresses.** (A) Profiles upon drought or salt treatment, where color scale represents FPKM normalized $\log_{10}$ transformed counts, where navy indicates low expression and firebrick3 indicates high expression. (B) $\log_{10}$ transformed fold-changes upon *Colletotrichum* infection and BA/GA treatments.

(*Amborella Genome Project, 2013*). RBOHN, which is absent from arabidopsis and rice, can be found in several dicots as well as monocots, e.g., cassava, rubber, cacao (*Theobroma cacao*), cucumber (*Cucumis sativus*), *Aquilegia coerulea*, banana (*Musa acuminata*), maize (*Zea mays*), and sorghum (*Sorghum bicolor*) (Table S3), supporting species/lineage-specific gene loss of this special group. Both RBOHB and RBOHF are widely found in angiosperms but not in *A. trichopoda*, suggesting that they may appear sometime before monocot-dicot divergence. RBOHC seems to be the youngest one that is widely present in eudicots but not in *A. coerulea*, a basal eudicot plant (*Zou et al., 2018*). Thereby, RBOHC may result from the γ WGD shared by core eudicots (*Jiao et al., 2012*). In contrast to jatropha and castor bean that harbor a single member for all seven phylogenetic groups, lineage-specific gain or loss of *Rboh* family genes was frequently observed in other species, especially those having experienced recent WGDs. For example, four paralogs present in arabidopsis were shown to result from β WGD (1), α WGD (2) as well as local duplication (1); one grape duplicate was derived from local duplication; four rice duplicates were all derived from WGD shared by Gramineous plants such as maize, sorghum, millet (*Setaria viridis*), and foxtail (*Setaria italica*); two cassava and three rubber duplicates were all derived from the ρ WGD shared by these two species; and, lineage-specific expansion was also observed in banana (*Musa acuminata*), poplar (*Populus trichocarpa*), and Brassicaceous plants such as *Arabidopsis lyrata*, *Brassica oleracea*, and *B. rapa* (Tables S1 and S3).

Although exhibiting one-to-one orthologous relationship and a high level of syntenic relation, divergence of several *Rboh* genes was observed between jatropha and castor bean. Loss of certain introns was observed for *JcRbohB*, *JcRbohD*, and *RcRbohN*, though the frequency is relatively lower than that in arabidopsis and rice. Nevertheless, coding

sequences are highly conserved and all deduced proteins were shown to contain typical domains essential for superoxide production, though Jc- and RcRBOHN feature a relative shorter N-terminal (Fig. 3).

Potential roles of *Jc/RcRbohs* could be inferred from their expression patterns and function-characterized orthologs in arabidopsis, rice, and other species. According to GO annotation, Rbohs have calcium ion binding, flavin adenine dinucleotide binding, protein binding and superoxide-generating NADPH oxidase activity, and are mainly located to plasma membrane/vacuole that are involved in various biological process as summarized in Table S4. Like *RcRbohC* and *RcRbohF* (see Fig. S2), *JcRbohC* and *JcRbohD* are ubiquitously expressed in most examined tissues of jatropha, supporting their key roles in these tissues. Similar expression patterns were also reported for *AtRbohD*, *AtRbohF*, *VvRbohA*, *VvRbohD*, *OsRbohB*, *OsRbohA*, *OsRbohC*, *OsRbohF*, *HbRbohC1*, *HbRbohF2*, *MeRbohC1*, and *MeRbohF2* in arabidopsis, rice, grape, cassava, and rubber (*Sagi & Fluhr, 2006*; *Cheng et al., 2013*; *Wang et al., 2013*; *Zou & Yang, 2019a*). Interestingly, *JcRbohB* was highly expressed in developing seeds, however, the transcript level of *RcRbohB* was shown to be extremely low in seeds, regardless of developing or germinating seeds (Fig. S2), supporting their possible function divergence. As reported for *AtRbohH* and *AtRbohJ* (*Sagi & Fluhr, 2006*), both *RcRbohH* and *JcRbohH* are preferentially expressed in male flowers (Fig. 4 and Fig. S2), implying their similar functions. Moreover, the involvement of *JcRbohs* in flower development and stress responses was also observed. *JcRbohB*, *JcRbohC*, *JcRbohD*, and *JcRbohN* are highly expressed in five stages of flower development, where *JcRbohB*, *JcRbohC*, and *JcRbohD* were shown be significantly regulated between at least two stages. Additionally, *JcRbohN*, *JcRbohD*, and *JcRbohF* are associated with pathogenic infection, where *JcRbohN* is also involved in BA response. In developing castor bean seeds cultured *in vitro*, transcriptome profiling also showed that *RcRbohE* and *RcRbohH* were significantly downregulated by ABA (*Chandrasekaran, Xu & Liu, 2014*). Thus far, the involvement in pathogenic infection and stress responses has been reported for various *Rboh* genes in other species, including *AtRbohD*, *AtRbohF*, *AtRbohJ*, *OsRbohF1*, *OsRbohH1*, *OsRbohE1*, *VvRbohD1*, *VvrbohD2*, *VvRbohF*, *VvRbohB*, *VvRbohC*, *VvRbohH*, *HbRbohB*, *HbRbohC1*, *HbRbohC2*, *HbRbohD1*, *HbRbohD2*, *HbRbohF1*, *HbRbohF2*, and *HbRbohN* (*Evans et al., 2005*; *Sagi & Fluhr, 2006*; *Maruta et al., 2011*; *Xie et al., 2011*; *Chaouch, Queval & Noctor, 2012*; *Cheng et al., 2013*; *Wang et al., 2013*; *Kaur et al., 2014*; *Chang et al., 2016*; *Zou & Yang, 2019a*). More recently, the RBOHF2 in barley was shown to be involved in salicylic acid accumulation and powdery mildew resistance (*Torres et al., 2017*).

## CONCLUSION

To the best of our knowledge, this is the first genome-wide comparative evolutionary analysis of *Rboh* family genes in jatropha and castor bean. The family number of seven members is relatively smaller than that reported in most angiosperm plants, reflecting no recent WGD occurred in these two special species. Nevertheless, the family is highly diverse, and seven phylogenetic or orthologous groups were found. Among them, RBOHN, a novel but ancient group, was shown to have been lost in many lineages. In jatropha, *JcRbohN*

was shown to widely participate in flower development, hormone signaling, pathogenic infection, and various abiotic stress responses. Additionally, conserved synteny and one-to-one orthologous relationship were observed between *JcRbohs* and *RcRbohs*, though gene-specific loss of certain introns was observed for several members. These findings will not only improve our knowledge on species-specific evolution of the *Rboh* gene family, but also facilitate further functional analysis of *Rboh* genes in jatropha and species beyond.

## ACKNOWLEDGEMENTS

The authors appreciate those contributors who make the related genome and transcriptome data accessible in public databases. They also thank the editor and two reviewers for their helpful suggestions.

### Funding

This work was supported by the National Natural Science Foundation of China (31700580), the Natural Science Foundation of Hainan province (319MS093), the Central Public-interest Scientific Institution Basal Research Fund for Chinese Academy of Tropical Agricultural Sciences (1630052017011 and 1630022019017), and the Research Fund of Guangdong University of Petrochemical Technology (2018rc55). The funders had no role in study design, data collection and analysis, decision to publish, or preparation of the manuscript.

### Grant Disclosures

The following grant information was disclosed by the authors:
National Natural Science Foundation of China: 31700580.
Natural Science Foundation of Hainan province: 319MS093.
Central Public-interest Scientific Institution Basal Research Fund for Chinese Academy of Tropical Agricultural Sciences: 1630052017011, 1630022019017.
Research Fund of Guangdong University of Petrochemical Technology: 2018rc55.

### Competing Interests

The authors declare there are no competing interests.

### Author Contributions

- Yongguo Zhao performed the experiments, analyzed the data, approved the final draft.
- Zhi Zou conceived and designed the experiments, performed the experiments, analyzed the data, contributed reagents/materials/analysis tools, prepared figures and/or tables, authored or reviewed drafts of the paper, approved the final draft.

### DNA Deposition

The following information was supplied regarding the deposition of DNA sequences:
 All the sequences obtained in this study available at NCBI GenBank under accession numbers as follows: *JcRbohB* (MK896910), *JcRbohC* (MK896909), *JcRbohD*

(MK896908), *JcRbohE* (MK896911), *JcRbohF* (MK896912), *JcRbohH* (MK896914), *JcRbohN* (MK896913), *RcRbohB* (MK896915), *RcRbohC* (MK896916), *RcRbohD* (MK896917), *RcRbohE* (MK896921), *RcRbohF* (MK896918), *RcRbohH* (MK896920), *RcRbohN* (MK896919).

## Data Availability

The accession numbers of the gene sequences are available in Table S1.

## Supplemental Information

Supplemental information for this article can be found online at http://dx.doi.org/10.7717/peerj.7263#supplemental-information.

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
