# Peer review of "Genomics analysis of genes encoding respiratory burst oxidase homologs (RBOHs) in jatropha and the comparison with castor bean"

_PeerJ, doi:10.7717/peerj.7263_

## Round 0.1 · original submission · Major Revisions

Dear author

Your paper has been assessed by three reviewers and myself as academic Editor.

As you could see below, the manuscript needs a major revision.
Most importantly:

Please clarify the methods. Add experiments of gene expression analysis
Misconceptions need to be corrected.
Statistical treatment of data needs to be more rigorous.
Please address all concerns of the reviewers and submit a revised version of the manuscript. Please include a detailed response to each reviewer.

I have been informed that PeerJ is now offering a language correction service for a fee. Please consider it before submitting the new version of the manuscript.

·

Basic reporting

Abstract
Lines 36-38. Phrase is not clear.. no duplicate was identified in both.. corresponding to no recente whole... is a double negation..

M&M section
Line 90.. Arabidopsis and rice rboh genes were obtained ... that means this genes were amplified?, cloned?

Line 114..Gene expression analysis..
the gene expression was done in silico?,
under what conditions the plants were grown?

Results section
Line 133-134.. seems repetitive to 136-141.
Sube 166. and Isrbigd, And variation..

Lines 170-181,
after describir fig 2a and 2b, jumps to figure 3,
then Lines 182-188, is describing figure 2c,

Line 172. The average theoretical MW (molecular weight)
Line 174. average pI .. the I is the italic letter

Expression profiles jcrboh genes... what about castor genes?
this section is just in silico analysis?,
but conditions of experiments could change from experiment to experiment.? so you do not have control of expression profile?
IND, STD, PID are not defiende.

Line 214.. the response to drought and salt stress was investigase....
this experiment was carried out in the present work or was also in silico analysis?,
if was done in this work, information about the experimentos should be added in M&M section.

Line 227.. respondes to coletotricum.. was investigated.. no mention in M&M section
Line 233.. regulación by hormones.. not mention in M&M section.

Figures
Fig 2A. the three is mixing C,G, A sequences, H, B, sequences,
seems not a very good classification of groups
Describe in figure legend what means motif 1, motif 2 and so on

Fig 4. describe in figure legend the meaning of std, , pid, ind,
leaf age¿, which age?,

Experimental design

Needs to be more clear,
the work was an in silico analysis? experiments of gene expression analysis are not described

Validity of the findings

there is not a results supported by experimental design explained in the text.

Additional comments

The most concern is about the gene expression analysis,
no information about the experiments design are present,
then is not clear if presented data is also an in silico analysis reviewing all information deposited at databases.

Reviewer 2 ·

Basic reporting

The MS is well written and presented. Please notice that R. communis common name id “castor bean” (not just “castor”).

Experimental design

Since data for expression analysis were taken from public data banks,how did authors make sure they were comparable? At present form, I'm not totally convinced that differences noticed, both among tissues and conditions are really comparable and, therefore, representative. It would be advisable to confirm these trends with actual lab results (PCR analysis of transcripts abundance, for example).

Validity of the findings

See comments in the previous section.

Additional comments

My only reason to rccomend minor revisions appears above.

Reviewer 3 ·

Basic reporting

The manuscript “Genomics analysis of the respiratory burst oxidase homolog (Rboh) gene family in jatropha and the comparison with castor” attempts to presents a genome-wide comparative analysis of Rboh family genes in economically important non-food oilseed crop jatropha (Jatropha curcas) and castor (Ricinus communis). Overall, the manuscript conforms to the structure recommended by the journal viz. Introduction, Materials and Methods, Results, Discussion, and Conclusions. The findings of the study are well discussed and supported by existing evidence. The overall English comprehension and grammar of the article is good and attains standards of a scientific article. The phylogenetic analysis of jatropha and castor with model monocot and dicot provide insights into species-specific functions. However, below are a few suggestions that would be useful for the readers and make it scientifically thorough.

1. Supplementary table S1 and S2 can be combined into a single excel file with information of S1 and S2 table as two separate sheets.
2. Please improve the legend of Fig 1 by incorporating information for e.g. the scale of the bars etc.
3. In the legend of Fig. 2, please provide the scale for the number of amino acid substitutions per site. Also, mention in the text what do the numeric values represented on the top of each clade correspond to?

Experimental design

1. In line 116, how did the author obtain clear reads from raw reads? Please specify exclusively in text.
2. Usually, the average width of the transcription factor binding site is 10bp. Why did the author select such a wide range from six to 180 residues in line 113? Please justify.
3. Please specify Bowtie2 parameters in line 117, unless used default.
4. Please specify if authors have used either existing packages or in-house scripts for identification of differentially expressed genes (DEG) in line 120-121.
5. In addition to DEGs, it would be great to provide a list of all the genes with expression and significance as supplementary.

Validity of the findings

The study is experimentally sound and well presented. For further comments refer section Experimental design.

---

## Round 0.2 · Minor Revisions

Dear authors

Gerard Lazo, the Section Editor wants to suggest the following:
“The manuscript reads well, but is lacking in updating the gene family characterization by using an ontology annotation, or by providing the new sequence data to a public resource (like GenBank) where it can be made available in proper formats. Some of this is in supplemental files, but none of these made their way towards being characterized in a public resource. Likewise, if an EST is mentioned the accession number, bioproject sequence, or assembly data should be pointed to so that sequence validation can be conducted by the reader. Journal manuscripts are often scanned by text-mining software that locates and extracts core data elements, like gene function. Adding standard ontology terms, such as the Gene Ontology (GO, geneontology.org) or others from the OBO foundry (obofoundry.org) can enhance the recognition of your contribution and description. This will also make human curation of literature easier and more accurate. None of this was visible. Since many bioinformatic exercises rely on easily adapted pipelines for sequence characterization, we should have such manuscripts add further characterization so that comparisons can readily be created using works already in the public domain. Adding useful annotations will go far in building informative databases. GO terms are a strong attribute tied in with the Arabidopsis and many other communities, and as these resources were cited it would be a complementing service to do the same, and would heighten the value of the manuscript. It is the use of such terms which have moved the technology further in recent years. As different expression profiles were addressed and the fine detail of the gene structure was also addressed, adding appropriate GO terms would be very helpful to the readership of this gene family. This should not be a large hurdle for the authors.“

Please follow the advice and submit a revised version of the manuscript.

·

Basic reporting

The manuscript was improved,
Line 168... And variety.. An variety?
Line 176...pI was not corrected as p(italics I)

Experimental design

no further comments

Validity of the findings

no further comments

Additional comments

Review some typos alongó the text

Reviewer 2 ·

Basic reporting

I consider that my comments were properly adressed, therefore the MS could be now published.

Experimental design

I consider that my comments were properly adressed, therefore the MS could be now published.

Validity of the findings

I consider that my comments were properly adressed, therefore the MS could be now published.

Additional comments

I consider that my comments were properly adressed, therefore the MS could be now published.

---

## Round 0.3 · accepted · Accept

Dear author

I can read that you have addressed the section editor concerns. The reviewers comments have been responded adequately and you have added the requested information on GO annotations.

I congratulate you for the nice piece of work, which will add value to PeerJ.